Evaluation of a new Argovit as an antiviral agent included in feed to protect the shrimp Litopenaeus vannamei against White Spot Syndrome Virus infection

Romo-Quiñonez Carlos R. 1
Álvarez-Sánchez Ana R. 2
Álvarez-Ruiz Pindaro 3
Chávez-Sánchez Maria C. 4
Bogdanchikova Nina 5
Pestryakov Alexey 6
Mejia-Ruiz Claudio H. hmejia04@cibnor.mx 1
1 Laboratorio Biotecnologia de Organismos Marinos, Programa de Acuicultura, Centro de Investigaciones Biológicas del Noroeste , La Paz , BCS , México
2 Universidad Técnica Estatal de Quevedo , Quevedo , Los Rios , Ecuador
3 Departamento de Acuicultura, Centro Interdisciplinario de Investigación para el Desarrollo Integral Regional, I.P.N. , Guasave , Sinaloa , México
4 Unidad de Acuicultura y Manejo Ambiental, Centro de Investigación en Alimentación y Desarrollo , Mazatlan , Sinaloa , México
5 Centro de Nanociencias y Nanotecnología, Universidad Nacional Autonoma de Mexico , Ensenada , Baja California , México
6 Tomsk Polytechnical University , Tomsk , Russia
Ford Alex
Electronic publication date: 2020 Feb 27
Publication date: 2020
Volume: 8
Electronic Location ID: e8446
Received 2019 Jul 29; Accepted 2019 Dec 20
Copyright: ©2020 Romo-Quiñonez et al.
Copyright year: 2020
Copyright holder: Romo-Quiñonez et al.
License: This is an open access article distributed under the terms of the Creative Commons Attribution License, which permits unrestricted use, distribution, reproduction and adaptation in any medium and for any purpose provided that it is properly attributed. For attribution, the original author(s), title, publication source (PeerJ) and either DOI or URL of the article must be cited.
License URL: https://creativecommons.org/licenses/by/4.0/

Keywords: Feed additive, AgNps, Shrimp, WSSV

Funding: National Council of Science and Technology (CONACyT), Mexico 258607 Tomsk Polytechnic University Competitiveness Enhancement Program project VIU-RSCBMT-65/2019 This study was supported by the National Council of Science and Technology (CONACyT), Mexico, through the grant No. 258607 & Tomsk Polytechnic University Competitiveness Enhancement Program project VIU-RSCBMT-65/2019. The funders had no role in study design, data collection and analysis, decision to publish, or preparation of the manuscript.

==============================
In this study, four experimental assays were conducted to evaluate the use of a new silver nanoparticle formulation named Argovit-4, which was prepared with slight modifications to enhance its biological activity against white spot syndrome virus (WSSV) in shrimp culture. The goals of these assays were to (1) determine the protective effect of Argovit-4 against WSSV, (2) determine whether Argovit-4 supplemented in feed exhibits toxicity towards shrimp, (3) determine whether Argovit-4 as antiviral additive in feed can prevent or delay/reduce WSSV-induced shrimp mortality, and (4) determine whether Argovit-4 supplemented in feed alters the early stages of the shrimp immune response. In bioassay 1, several viral inocula calibrated at 7 SID50(shrimp infectious doses 50% endpoint) were exposed to 40, 100, 200 and 1,000 ng/SID50 of Ag+ and then intramuscularly injected into shrimp for 96 h. In bioassay 2, shrimp were fed Argovit-4 supplemented in feed at different concentrations (10, 100 and 1,000 µg per gram of feed) for 192 h. In bioassay 3, shrimp were treated with Argovit-4 supplemented in feed at different concentrations and then challenged against WSSV for 192 h. In bioassay 4, quantitative real-time RT-qPCR was performed to measure the transcriptional responses of five immune-relevant genes in haemocytes of experimental shrimp treated with Argovit-4 supplemented in feed at 0, 6, 12, 24 and 48 h. The intramuscularly injected Argovit-4 showed a dose-dependent effect (p < 0.05) on the cumulative shrimp mortality from 0–96 h post-infection. In the second bioassay, shrimp fed Argovit-4 supplemented in feed did not show signs of toxicity for the assayed doses over the 192-h experiment. The third and fourth bioassays showed that shrimp challenged with WSSV at 1,000 µg/g feed exhibited reduced mortality without altering the expression of some immune system-related genes according to the observed level of transcriptional. This study is the first show that the new Argovit-4 formulation has potential as an antiviral additive in feed against WSSV and demonstrates a practical therapeutic strategy to control WSSV and possibly other invertebrate pathogens in shrimp aquaculture.

Introduction

The white spot syndrome virus (WSSV) is highly aggressive towards shrimp aquaculture worldwide, for which several studies have attempted to develop a treatment (Álvarez Sánchez et al., 2018). This pathogen can cause 100% mortality in shrimp culture in just a few days, causing large economic losses, business closures and high unemployment rates (Sanchez-Martínez, Aguirre-Guzman & Mejía-Ruíz, 2007; Bustillo, Escobedo & Sotelo, 2009). Currently, nanotechnology has attracted a great deal of attention as an emerging field of research and technological development that open the possibility of handling material at the nanometric level, with dimensions between 1 and 100 nm (Meneses-Márquez et al., 2018). In this size range, nanoparticles show new physical and chemical properties that can be used in different scientific areas for the unprecedented study of phenomena that occur at the atomic and molecular level (Parveen, Misra & Sahoo, 2012; Esmaeillou et al., 2017). Among the most interesting and promising nanomaterials are silver nanoparticles (AgNPs), which have shown broad-spectrum antiviral activity (Bello-Bello et al., 2018; Galdiero et al., 2011; Chris, Singh & Agarwal, 2018). Several in vitro studies have demonstrated the effect of AgNPs as an antiviral alternative against human viruses, including human immunodeficiency virus (Elechiguerra et al., 2005; Lara et al., 2010), H1N1 influenza A virus (Mori et al., 2013), monkeypox virus (Rogers et al., 2008), Tacaribe virus (Speshock et al., 2010), hepatitis B virus (Lu et al., 2008) and herpes simplex virus (Baram-Pinto et al., 2009).

The mechanism of action of AgNPs as an antiviral agent has been studied against several enveloped viruses (Lara et al., 2011; Bogdanchikova et al., 2016), and it has suggested. that AgNPs undergo preferential binding with viral envelope glycoproteins to inhibit viruses from binding to host cells (Elechiguerra et al., 2005; Lara et al., 2010). In addition, multiple studies have shown that nanoparticles can be recognized by the immune system and modulate the induction of immunostimulatory effects (Dacoba et al., 2017; Boraschi et al., 2017).

Recent advances have been made in aquaculture using AgNP-based formulations. In one study, Argovit® that was administered intramuscularly in shrimp Litopenaeus vannamei showed antiviral activity against WSSV (Juárez-Moreno et al., 2017), a highly lethal and contagious pathogen. This result indicated that a single dose of silver nanoparticles was able to increase shrimp survival by up to 80% without toxic effects. In another study, Ochoa-Meza et al. (2019) evaluated the ability of silver nanoparticles coated with commercial non-toxic PVP (Argovit®) to promote the immune system response of shrimp infected with WSSV and observed a late immunostimulatory activity of two immune-related genes. In the same report, authors showed that a single dose of AgNP was able to increase the survival of the shrimp against severe infections of WSSV, even under adverse conditions in the presence of a high concentration of ferrous ions.

Crustaceans have an innate immune system, similar to other invertebrates; however, they lack a developed adaptive immune system. The innate immune system of shrimp consists of several natural defence mechanisms: a microbial recognition system, a prophenoloxidase system (proPO system), a coagulation system, phagocytosis, encapsulation and the formation of nodules and oxygen reactive compounds (Kurtz, 2005; Ji, Yao & Wang, 2009; Hirono et al., 2011). Several genes that participate in the shrimp immune system have been described in the species L. vannamei, some of which have been described in previous reports from our group (Flegel & Sritunyalucksana, 2010). To identify a potential mechanism of the antiviral and immunostimulatory activities of silver nanoparticles in shrimp, we selected five genes to study that are representative of different immunological routes. Two of the genes encode proteins (Rab6 and PAP) that participate in endocytosis and phagocytosis (Ye, Tang & Zhang, 2012; Khimmakthong et al., 2013). The genes encoding Crustin and PEN4 have been identified as participating in antiviral immunity against WSSV infections in L. vannamei (Sun, Wang & Zhu, 2017; Huan et al., 2011). Finally, the proPO gene activates the proPO system, which is involved in melanin formation in arthropods and other invertebrates (Sritunyalucksana & Soderhall, 2000) and results in the production of the melanin pigment that can often be seen as dark spots in the cuticle of arthropods and two antimicrobial peptides from defence system also in WSSV infections (Abad-Rosales et al., 2019). These advances have demonstrated that AgNPs are effective against a wide spectrum of viruses, especially against the disease-causing WSSV, for which there have been no effective treatments until now (Verbruggen et al., 2016). However, intramuscular administration of AgNPs is impractical, considering the large volumes of organisms to that would need to be treated. A practical alternative is the inclusion of AgNPs in balanced shrimp feed (Dananjaya et al., 2016), guaranteeing a greater usefulness of AgNPs therapy.

Although the Argovit® formulation is no longer commercially available, new formulations have been developed with unique features (N Bogdanchikova, Pers. comm., 2017). For instance, the Argovit-4 formulation retains essentially same the silver nanoparticle design of Argovit®. Thus, in this study, we investigated whether the AgNPs-based Argovit-4 could be used as an antiviral additive in feed to interfere with WSSV infection in shrimp.

Materials and Methods

Argovit-4 formulation

The silver nanoparticles Argovit-4, which replaced Argovit® (Novosibirsk, Russia; patent 2427380) was kindly donated by Dr. Nina Bogdanchikova from CNyN-UNAM, and this can be requested by contacting Dr. Bodganchikova, (or it can be purchased at Bionag SAPI de CV, Tijuana, México. MC Martha Alvarado, (52) 664-317-99-21). The supplier’s specifications indicate that Argovit-4 consists of spherical AgNPs with an average size of 35 ± 15 nm that are functionalized with non-toxic PVP (K30) in water. Table 1, which was modified from Juárez-Moreno et al. (2017), summarizes the physicochemical characterization of these formulations.

Table 1 Physicochemical Characteristics of silver nanoparticles.

Physicochemical characteristics of silver nanoparticles modied to Argovit-4 from Juárez-Moreno et al. (2017).

Properties	Mean	
K-value	K30	
PVP content	18.8 (% wt.)	
Metallic silver content	1.2 (% wt.)	
Silver nanoparticle morphology	spheroidal	
Average diameter of metallic silver particles by TEM
data (nm)	35 ± 15	
Size interval of metallic silver particles by TEM data	1 to 90 (nm)	
Hydrodynamic diameter: metallic Ag with PVP	70 (nm)	
Zeta potential	−15 (mV)	
Surface plasmon resonance	420 (nm)	
PVP structure by FTIR	Confirmed	

White spot syndrome viral stock

The WSSV strain (Guasave isolate, 2005) used in this study was kindly provided by the Biochemistry and Molecular Biology Laboratory from CIIDIR, Sinaloa, Mexico. The viral inoculum was prepared according to the methodology described by Álvarez Ruiz et al. (2013). Muscular tissues from moribund shrimp were collected, weighed and homogenized in 10 volumes (W/V) of phosphate-buffered saline (1 × PBS; 137 mM NaCl, 2.7 mM KCl, 10 mM Na2HPO4, and 2 mM KH2PO4) to pH 7.4. The homogenized tissue was centrifuged at 13,000× g (or g-force) for 20 min at 4 °C, after which the supernatant was transferred to another microtube (1.6 mL) and centrifuged again at 13,000× g for 20 min at 4 °C. The supernatant was passed through a 0.45 µm pore-sized filter and stored at −70 °C.

Preparation of inoculum from WSSV-infected tissue

The experimental challenge with WSSV was conducted by the oral route using infected tissue. The infected tissue was prepared by intramuscularly injecting a batch of shrimp (n = 10) of 10 ± 2 g with of 32 SID50 in 100 µL of PBS (assuming that 100% of organisms would be infected at 54 h post-infection). The shrimp with observable clinical signs of WSSV infection were individually separated and stored at −70 °C until all of the shrimp became infected. Subsequently, after removing the cephalothorax, pleopods and exoskeleton the muscle tissue of the dead WSSV-infected shrimp was weighed and homogenized in 120 mL of marine water for every 24 g of tissue. The homogenized material was used as the inoculum used for the bioassay 4.

Shrimps

Batches of L. vannamei (4 ± 0.5 g) juveniles were obtained from a local farm in Guasave, Sinaloa, Mexico. The shrimps were transported to CIIDIR-IPN and placed in plastic tanks (1000-L capacity) with 500 L of filtered seawater at 30 PSU (Practical Salinity Units) with constant aeration. The shrimps were fed ad libitum twice daily with Camaronina® commercial feed (Purina-Cargill LTD, Minnesota, USA) with 35% protein. Subsequently, the shrimps were verified to be free of WSSV, IHHNV and AHPND by endpoint PCR.

Experimental conditions

All experiments were performed in glass tanks (40-L capacity) containing 25 L of seawater (30 PSU) at 27 ± 1 °C and continuous aeration with the shrimps acclimatized under these conditions for 24 h before starting the experiments. The experimental shrimp for bioassay 1 and 2 were fed Camaronina® at 4% of their body weight, while in bioassays 3, 4 and 5, the shrimp were fed feed supplemented with AgNPs. The uneaten feed and faecal material were removed by siphoning every 2 days with water exchange (50% volume) during all experimental assays.

In vivo viral titration and determination of minimal infectious dose by the intramuscular route

A batch of shrimp was used to titre the fresh inoculum using the intramuscular injection method described previously by Álvarez Ruiz et al. (2013) and Apun et al. (2017). The inoculum was titrated by in vivo injecting 100 µL of serial dilutions from the original extract (10−3, 10−4, 10−5 and 10−6) between the 3rd and 4th abdominal segments of shrimp (n = 5 shrimp per dilution). The experimental assay was finished at 96 h post-infection (hpi). The viral titre was calculated using the method of Reed & Muench (1938), for the concentration of a test substance that produces an effect of interest in half of the test units (50% tissue culture infectious dose of a virus), which was described as SID5 0/mL.

Preparation of Argovit-4 supplemented feed

Feed supplemented with AgNPs at different concentrations was made according to the modified method of Gutiérrez-Dagnino et al. (2015). Camaronina® 35% protein pellets (commercial feed for shrimp) were pulverized in a coffee grinder. Three homogeneous pastes supplemented with Argovit-4 AgNPs (10, 100 and 1,000 µg/g feed), sodium alginate (20 µg/g feed; Sigma-Aldrich®, St. Louis, MO, USA), and distilled water (0.5 mL/g feed) were generated. In the control feed, AgNPs were replaced by distilled water. Subsequently, the pastes were manually pelletized with a 50-mL disposable syringe (Millipore®, Merck, Darmstadt, Germany) and dried for 72 h at 4 °C in darkness.

Bioassays

Bioassay 1: determination of the minimum WSSV infectious dose by intramuscular route

Bioassay 1 was designed to determine the effective dose of WSSV that is capable of infecting 100% of the shrimp (as determined by conspicuous clinical signs, such as a lack of appetite, erratic swimming and conspicuous atypical coloration). Three groups of shrimp (30 organisms per group, n = 10 shrimp in three replicates) were intramuscularly inoculated between the 3rd and 4th abdominal segment with different WSSV infectious doses (7, 10 and 100 SID50 in l00 µL of PBS). The control group was a mock inoculation with 100 µL of PBS. The mortality of the shrimp in every group was recorded and the bioassay finished was when all of the infected shrimp died.

Bioassay 2: inhibition of pathogenicity by the application of a WSSV-Argovit-4 mixture

Several viral inoculums were calibrated at 7 SID50 in 100 µL volumes and exposed at 40, 100, 200 and 1,000 ng of Argovit-4. The viral inoculums and Argovit-4 (WSSV + AgNPs) mixtures were incubated for one hour on ice and then intramuscularly injected into 30 shrimp for every group (n = 10 organisms per replicate) for each treatment. The positive control group was inoculated with WSSV (without AgNPs), while the control group was mock-treated with 100 µL of PBS. The experiment lasted up to 96 hpi, and every six hours the presence of dead organisms, clinical signs and the behaviour of the surviving shrimp were recorded.

Bioassay 3: toxic effect of Argovit-4 supplemented in feed

The toxic effect of Argovit-4 supplemented in feed at different concentrations was tested first. The experiment consisted of four treatments with three replicates each (n = 10 shrimp per replicate) that were organized as follows: (1) shrimps receiving feed without AgNPs (control group), (2) shrimps receiving AgNPs at 10 µg/g feed, (3) shrimps receiving AgNPs at 100 µg/g feed, (4) shrimps receiving AgNPs at 1,000 µg/g feed. The shrimps were fed twice daily at 4% of their body weight. The signs of toxicity (lethargic movement, empty gut or atypical colour change) and cumulative mortality were recorded daily. Previously a toxicity bioassay was carried out by intramuscular route in the shrimp for each of 5 different designs of Argovit, which did not occurred mortality and Argovit-4 was selected (data not shown).

Bioassay 4: argovit-4 supplemented in feed against WSSV

For bioassay 4, the experimental design was performed in triplicate (n = 10 shrimps by replicate). The shrimp were fed Argovit-4-supplemented feed at 0 and 12 h for the first and second feedings, respectively. In both cases, the feed weight corresponded to 4% of the shrimp body weight (4 g/organism). Subsequently, at 24 h, the shrimps were fed the WSSV inoculum (200 mg infected tissue per shrimp). Every shrimp received two daily rations of food corresponding to 4% of their body weight, and at 12 h post-treatment with AgNP in feed were challenged with 200 mg of infected tissue per shrimp for each replicate. The cumulative mortality was recorded twice a day until 192 h.

Bioassay 5: immunostimulatory activity of Argovit-4 towards immune-related genes

To determine the immunostimulatory activity of Argovit-4 on shrimp L. vannamei juveniles, a bioassay was established in two glass tanks (40 L) with 25 L of seawater and continuous aeration. Two groups (n = 12 shrimp per tank) were treated with either feed supplemented with Argovit-4 (1,000 µg/g) or control feed (feed without Argovit-4) at 4% of their body weight twice daily.

Haemolymph sampling

Shrimp haemolymph samples from each tank were collected in pools of three to obtain ∼400 µL/shrimp every 0, 6, 12 and 48 h post-treatment. A one mL syringe (27 G ×13 mm needle) was used for rinsing with 5% potassium oxalate in isotonic saline (850 mOsm/kg) to obtain haemolymph from the ventral sinus (base of the first abdominal segment Apun-Molina et al., 2017; Guemez-Sorhouet et al., 2019). The pools of haemolymph samples were centrifuged at 800× g for 10 min at 4 °C, after which the plasma was removed and the pellet was washed with 250 µL of cold PBS centrifugation as described above. The supernatant was removed, and the haemocytes were suspended in microtubes containing 200 µL of precooled TRIzol Reagent (TRI Reagent®, Molecular Research Center, Ohio, USA). The haemocyte samples were stored at −70 °C until use.

Isolation of RNA and cDNA synthesis

Total RNA from the haemocyte and intestinal samples was extracted with TRIzol reagent according to the manufacturer’s protocol. The RNA was quantified at 260 nm using a NanoDrop instrument (NanoDrop Technologies, Wilmington, DE) and then treated with DNase 1 (1 U/µL, Sigma-Aldrich®, St. Louis, MO, USA) to remove residual genomic DNA. cDNA synthesis was performed using the Improm II® Reverse Transcription System (Promega, Wisconsin, USA) with 500 ng of total RNA and random primers from the kit. The obtained cDNA was suspended in 80 µL of ultrapure water and stored at −70 °C until analysis.

Quantitative real-time PCR analysis

The transcriptional responses of genes encoding immune-related proteins (PEN4, Crustin, proPO, PAP and Rab6) from L. vannamei haemocytes were determined by quantitative real-time PCR (RT-qPCR) using a CFX96 system and CFX Manager version 3.0 (Bio-Rad Laboratories, Hercules, CA, USA). The expression of target genes was normalized to that of the β-actin gene as an endogenous control. Table 2 shows the sequences of specific primers used in this study. All amplifications were performed (in duplicate) in 15 µL volume reactions and contained 7.5 µL of PCR master-mix 2 × (3.0 µL of 5 × Colorless GoTaq Flexi Buffer [Promega, Wisconsin, USA], 1.5 µL of 25 mM MgCl2 [Promega], 0.3 µL of 10 mM dNTPs [Bioline, Taunton, MA, USA], 0.75 µL of 20 × EvaGreen® [Biotium, Fremont, CA, USA], 0.1 µL of 5 U/ µL GoTaq® Flexi DNA Polymerase [Promega, Wisconsin, USA], and 1.85 µL of ultrapure water). Each master-mix aliquot was mixed with 0.35 µL of each primer (10 µM, Sigma-Aldrich®), 1.8 µL of ultrapure water and 5 µL of cDNA as template. The thermocycling program was as follows: an initial cycle of 95 °C for 3 min, followed by 40 cycles of 95 °C for 10 s, 58 °C for 20 s, 72 °C for 30 s and 79 °C for 5 s (to acquire the fluorescence signal). After each reaction, a dissociation curve from 65 to 90 °C was recorded at 0.5 °C increments and examined for unique and specific products. The Pfaffl method was used to analyse the relative quantification data from the real-time PCR experiments (Pfaffl, 2003).

Table 2 Specic primers used.

Specic primers used for PCR amplications of immune system-related genes.

Genes	Primer sequences (5′–3′)	Amplification size (bp)	Reference	
ProPO	Fw: CTGGGCCCGGGAACTCAAG
Rv: GGTGAGCATGAAGAAGAGCTGGA	125	Soto-Alcalá et al. (2019)	
PEN 4	Fw: GCCCGTTACCCAAACCATC
Rv: CCGTATCTGAAGCAGCAAAGTC	106	Wang et al. (2010)	
PAP	Fw: CGAAGTTCAGGTTGTGCGTG
Rv: ACTGATGCACCATTGGCCTT	126	Soto-Alcalá et al. (2019)	
Rab 6	Fw: GTTCCGCAGCCTTATTCCCT
Rv: ATCACTGCCTCGCTCTGTTC	133	Soto-Alcalá et al. (2019)	
Crustin	Fw: GAGGGTCAAGCCTACTGCTG
Rv: ACTTATCGAGGCCAGCACAC	157	Wang et al. (2010)	
β-Actin	Fw: CCACGAGACCACCTACAAC
Rv: AGCGAGGGCAGTGATTTC	142	Wang, Chang & Chen (2007)	

Statistical analysis

The percentage data of cumulative mortality (Bioassays 2, 3 and 4) were initially arcsine-transformed to perform the statistical analysis. Normality and homogeneity of variances assumptions were analysed by Kolmogorov–Smirnov and Levene’s tests. Thereafter, Kruskal–Wallis ANOVA test (non-parametric ANOVA) followed by a multiple comparison test was used to determine significant differences between the experimental groups (p < 0.05).

To determine the effect of Argovit-4 on the transcriptional response of five immune-relevant genes (Bioassay 5), the data were Log10-transformed to confirm its normality before statistical analysis. The data were analysed by Student’s t-test to determine significant differences (p < 0.05). All bioassays were performed in triplicate and the results of cumulative mortality were expressed as mean ± standard deviation. While, the immune-related gene expression results are reported as the mean values ± standard error. Statistical analyses were performed using STATISTICA version 8.0 (StatSoft, Tulsa, OK).

Results

A 7 SID50 inoculum is the minimum effective dose to generate 100% shrimp mortality

In the challenge bioassay, the three assayed doses of WSSV resulted in 100% shrimp mortality at different time intervals. The shrimp group infected with 100 SID50 showed mortality after 30 hpi with values above 3%, reaching 100% of mortality at 54 hpi. In contrast, the shrimp group infected with 10 SID50 exhibited mortality after 36 hpi with values above 17%, reaching 100% at 60 hpi. Finally, in the shrimp group infected with 7 SID50, mortality began after 36 hpi with values above 13%, reaching 100% at 72 hpi. The control group (PBS) did not show mortality during the bioassay (Fig. 1). The results revealed that 7 SID50 as the minimal dose capable causing 100% of shrimp to be infected with conspicuous clinical signs (lack of appetite, erratic swimming and atypical coloration).

Figure 1 WSSV-Inoculum title.

Cumulative mortality of shrimp L. vannamei injected with three different WSSV doses by intramuscular route. Shrimps infected with 100, 10 and 7 SID. (f) Control negative shrimp treated with PBS. The results are presented as the mean ± SD.

Argovit-4/WSSV decreased the mortality of shrimp

The shrimp group treated with 1,000 ng/SID50 showed the lowest mortality (50%) at 60 hpi, whereas the shrimp groups treated with 40, 100 and 200 ng/SID50 of Argovit-4 showed mortalities of 90, 83 and 70% at 60 hpi, respectively. The positive control showed 100% mortality at 72 hpi compared with 0% in the control group. Statistical analysis revealed that the susceptibility of the shrimp to WSSV mixed with Argovit-4 significantly decreased (Kruskal–Wallis test; p = 0.0169) in a dose-dependent manner across the groups (Fig. 2).

Argovit-4 supplemented in feed is non-toxic

None of the assayed Argovit-4 doses supplemented in feed caused significant shrimp mortality (Kruskal–Wallis test; p = 0.0001) and not affect the health or survival of shrimp, given that they did not show atypical behaviour (evaluated by observations for empty guts, lethargy, and erratic movement) by toxicity effects during the course of the bioassay at 192 h post-treatment.

Argovit-4 supplemented in feed has an antiviral effect against WSSV

The shrimp group exposed to infected WSSV tissue (without Argovit-4) showed a mortality of 43% at 84 hpi. The cumulative mortality in shrimp fed the 1,000 µg/g doses was 3% after infection with WSSV at 192 h, followed by 10 and 16% for the 100 and 10 µg/g doses, respectively. In contrast, the uninfected control group did not show mortality. Interestingly, statistical analysis (Kruskal–Wallis test; p = 0.0001) did not reveal a dose-dependent effect (Fig. 3). However, the results revealed an obvious antiviral effect with both doses assayed.

Figure 2 Effect of mixture Argovit-4/WSSV of different doses injected by intramuscular route.

Cumulative mortality of L. vannamei juveniles: 40, 100, 200, 1,000 ng/(7)SID. (Control −) shrimp treated with PBS and were not infected; (Control +) shrimp treated with PBS and infected with WSSV inoculums. Statistical differences are represented by different letters (p < 0.05). Points are presented as the mean ± SD.

Figure 3 Effect of Argovit-4 supplemented in feed at different doses on cumulative mortality of L. vannamei juveniles.

Shrimp nourished with Argovit-4 supplemented feed with concentration of silver: 10, 100, 1,000 µg/g. (Control +) shrimp fed only with WSSV inoculum. (Control −) shrimp fed with feed lacking Argovit-4 additive. Procedure sequence: at 0 h the feeding with Argovit-4 supplemented feed was executed for the first time; at 12 h feeding with Argovit-4 supplemented feed was performed for the second time and at 24 h feeding with WSSV inoculum (200 mg infected tissue per shrimp) was performed. After that, every 12 h the shrimps were fed with commercial feed. Statistical differences are represented by different letters (p < 0.05). The results are presented as the mean ± SD.

Argovit-4 does not induce an early immunostimulatory effect for immune-related genes

The expression analysis of genes encoding Crustin, PEN4, PAP and Rab6 showed no significant (p < 0.05) changes in the haemocytes of shrimp treated with Argovit-4 supplemented in feed with respect to the control (Figs. 4A, 4B; 5A, 5B). Interestingly, only the gene encoding proPO was downregulated at 48 h (t-student test; p = 0.024), which was possibly an indirect effect of low levels of AgNPs in the haemolymph (Fig. 6).

Figure 4 Relative expression of PEN4 and Crustin genes.

The graphic show the RT-qPCR of; (A) Relative expression of PEN4 gene with no significant differences between AgNP treatment (black box) and control (white box); (B) Relative expression of Crustin gene, also no significant differences between AgNP treatment (black box) and control sample (white box). The data of the statistical analysis are represented in mean ± standard error.

Figure 5 Relative expression of Rab6 and PAP genes.

The graphic show the RT-qPCR of; (A) Relative expression of Rab6 gene with no significant differences between AgNP treatment (black box) and control (white box); (B) Relative expression of PAP gene, also no significant differences between AgNP treatment (black box) and control sample (white box). The data of the statistical analysis are represented in mean ± standard error.

Figure 6 Relative expression of proPO gene.

The graphic show the RT-qPCR of the relative expression of proPO gene with no significant differences between AgNP treatment (black box) and control (white box) only at 48 hrs. The data of the statistical analysis are represented in mean ± standard error (Analysis test in text).

Discussion

Several studies have evaluated the effectiveness of AgNPs in aquatic diseases control because to their antimicrobial potential against bacteria, virus and parasites (Barakat, El-Sayed & Gohar, 2016; Morales-Covarrubias et al., 2016; Acedo-Valdez et al., 2017; Pimentel-Acosta et al., 2019). However, this is the first work that examine the efficacy of Argovit-4 (AgNPs-based formulation) as a potential antiviral additive in feed to protect the health of shrimp L. vannamei against WSSV.

In order to evaluate the protective effect of AgNPs on shrimp against WSSV, firstly, we determinate that 7 SID50 was the minimum effective dose to generate 100% shrimp mortality according to WSSV inoculum used (bioassay 1) and then, we preincubated infectious WSSV inoculums (calibrated to 7 SID50) with Argovit-4 at different concentrations, which were subsequently intramuscularly injected in shrimp (bioassay 2). Our results indicated that WSSV infectivity is drastically affected due to a delay in the onset of clinical signs and mortality in a manner dose-dependent (p < 0.05) (Fig. 2), compared with that observed in the control (WSSV without Argovit-4). Some authors have observed that silver nanoparticles interaction with envelope glycoproteins of virus can inhibit post-entry stages during replication cycle (Rogers et al., 2008; Lara et al., 2010; Borrego et al., 2016). Elechiguerra et al. (2005) suggested that AgNPs undergo specific interaction with HIV-1 due to their affinity to disulfide bonded regions (thiol groups provided in viral envelope protein domains). While, Lara et al. (2010) postulated that silver nanoparticles not only bind to the viral protein, but can also structurally modify it through the denaturing of disulfide-bound domain. Hence, we hypothesize that AgNPs could interact with WSSV envelope proteins (e.g., VP28; Tang et al., 2007) and block the specific binding to membrane proteins of shrimp host cells. This can be seen in our results of the ability of AgNPs to decrease residual infectivity of viral particles after 60 min of incubation, as described by Lara et al. (2010). Nevertheless, further research is needed to elucidate the antiviral mechanism of silver nanoparticles against WSSV.

Recently, Juárez-Moreno et al. (2017) reported that prophylactic doses 50 ng/g Argovit® (AgNPs-based formulation) administered by intramuscular route in infected shrimp with WSSV evinced a rate survival of 80% without toxic effects after 96 h post infection, while Ochoa-Meza et al. (2019) showed evidence that a single therapeutic dose 1.2 µg/g AgNPs can delay the onset of clinical signs and enhance the response of shrimp immune system without toxic effects in healthy shrimps. However, this results are not comparable with Juárez-Moreno et al. (2017) and Ochoa-Meza et al. (2019) in effectiveness, due to the different: (1) AgNP preparations used, (2) weights of the shrimp and, (3) routes of exposure; considering that intramuscular administration is impractical in aquaculture due to the enormous quantity of organisms that are need to be treated (Kumar & Roy, 2017).

The administration by feeding represents the most practical prophylactic or therapeutic method for in situ shrimp culture (Alvarez-Sanchez et al., 2017). Therefore, we prepared feed supplemented with Argovit-4 at 10, 100 and 1,000 µg/g doses and determined their effects on shrimp health or mortality (bioassay 3). Our results showed no obvious toxicity effects on the behavioural activity of shrimp in the form of erratic swimming, changes in feeding rate, atypical body colour body, and increased mortality (1 shrimp death) in all treatments by the oral route. The lack of mortality and clinical signs can be explained by the structural design of Argovit® nanoparticles having been approved by sanitary international organizations for their use in veterinary and medical applications (Vazquez-Muñoz, Avalos-Borja & Castro-Longoria, 2014; Bogdanchikova et al., 2016; Borrego et al., 2016).

Despite the potential of AgNPs as antimicrobial agent, in aquaculture industry, concerns remain regarding the potential negative impact to the marine and freshwater environment. However, to date, there have not been clear the in vivo toxicity of silver nanoparticles in marine environment, as the effects depend on AgNP properties (particle size, surface area, shape), concentration, AgNP colloidal stability, aggregation grade, sedimentation, and on the amount and species of products yielded from chemical interactions between AgNPs and other variables (Fabrega et al., 2011; Sharma et al., 2019; Pimentel-Acosta et al., 2019).

It is important to clarify that Argovit-4, given its recent design, still does not present toxicity reports with which this AgNP should be better studied in terms of its safety if it is intended to apply to animals for human consumption and in turn study bioaccumulation in the same shrimp and in the ecotoxicological environment. We have presented in this research some results of AgNP exposure considering only the acute toxic response, that is, in the short term. This is because the antiviral activities observed are a consequence of the application of a single dose. The main reason why a chronic toxicological study was not performed is precisely because of the observed antiviral effect. Fabrega et al. (2011), points out that invertebrates such as water flea Daphnia pulex, acute and chronic toxicity studies do not generate bioaccumulation, although other authors point out that in the event that AgNPs accumulate in the gills, they do not have harmful side effects (Xiang et al., 2019), however, this could dependent of administered doses (Monfared et al., 2015). It is important to mention that more experiments should be carried out to consider the possibility of applying this nanoparticular design in the field. For example, bioassays that generate enough information to evaluate the bioaccumulation of flora and fauna accompanying the bodies of water where the shrimp are raised.

To determine the protective properties of Argovit-4 supplemented in feed against WSSV, the results of bioassay 4 indicate that the 1,000 µg/g treatment provided the best protective effect in shrimp against the inoculum of WSSV by the oral route than by injection (Fig. 3). This result was probably due to the high bioavailability of the silver nanoparticles in the digestive tract, which allowed the AgNPs-virus interactions to improve and promote entry by clathrin-mediated receptor routes (Coutiño, Lagunes-Ávila & Arroyo-Helguera, 2017). This is the first study that describes the use of AgNPs as an antiviral additive in feed against WSSV. In recent years, the use of metallic nanoparticles in aquaculture has been increased, for example in fish. For Clarias gariepinus, selenium nanoparticles (SeNPs) at a 680 µg/g dose was observed to improve feeds by increasing the proportion of fish food nutrients that pass across the gut tissue (Chris, Singh & Agarwal, 2018). For Clarias batrachus, it was also shown that iron nanoparticles (FeNPs) at a 400 µg/g dose is sufficient to ensure the growth and health of fish (Akter et al., 2018).

Previous studies have demonstrated that because metallic nanoparticles are foreign elements to organisms, they can be recognized and internalized by immune system cells through different routes according to their size and shape (Jiang et al., 2008; Coutiño, Lagunes-Ávila & Arroyo-Helguera, 2017; Dacoba et al., 2017; Boraschi et al., 2017). To test this possibility, and to determine the immunostimulatory effect in the early stages by assessing the expression of immune-relevant genes, bioassay 5 was performed in which shrimp were fed Argovit-4 at a 1,000 µg/g dose due to it being the therapeutic dose that best-protected shrimp against WSSV. Five genes encoding immune-related proteins were examined at different time intervals in shrimp haemocytes (PAP, Rab6, Crustin, PEN4, and proPO) by RT-qPCR analysis. PAP (Phagocytosis-Activating Protein) and Rab6 have been shown to be associated with phagocytosis in shrimp immune response (Deachamag et al., 2006; Zhao, Jianga & Zhang, 2011; Ye, Zong & Zhang, 2012; Ye, Tang & Zhang, 2012; Soto-Alcalá et al., 2019). Crustin and PEN4 are antimicrobial peptides (AMPs) that participate in the humoural defence against pathogens or foreign particles (Wang et al., 2010; Hirono et al., 2011), while proPO is an important humoural effector in innate immunity, catalysing the upstream steps of melanization in shrimp and others invertebrates (Gollas-Galván, Hernández-López & Vargas-Albores, 1999; Wang et al., 2010; Amparyup, Charoensapsri & Tassanakajon, 2013). The expression of the genes encoding PAP, Rab6, Crustin, and PEN4 in the treated shrimps showed no significant differences in the time intervals proposed with respect to the control (feed without Argovit-4). The expression of the gene encoding phagocytosis activating protein was not activated in samples from the silver nanoparticle-treated shrimp compared to that observed in the control shrimp samples, suggesting that the size of the particles is important, since this parameter facilitates their incorporation and internalization in haemocytes. This observation is reinforced by the same observed behaviour of the Rab6 gene, which participates in the endocytic pathway (Soto-Alcala et al., 2019), since its expression was not altered at any time by the silver nanoparticles. On the other hand, peneaidins are a family of antibacterial proteins that have recently been shown to contribute to the arrest of WSSV-type viral particles. However, the expression of the PEN4 gene was not altered with respect to the time recorded from zero to 48 h, which indicates that the AgNPs do not interact with the effectors of this gene. The same pattern was observed for the expression of the gene encoding Crustin, which has been shown to be activated in the first hours of WSSV infection (Huan et al., 2011; Sun, Wang & Zhu, 2017). Taken together, the patterns of the four assayed genes encoding PAP, Rab6, PEN3 and Crustin strongly suggests that the antiviral effect of the nanoparticles occurs by a different pathway than antibacterial peptides or endocytic phagocytosis.

The proPO system is one of the most studied in arthropods, where particles generated by cell destruction and attacks by pathogenic organisms are coated by melanin via the oxidation of phenols by the enzyme phenoloxidase, which are subsequently secreted as melanized corpuscles (Amparyup, Charoensapsri & Tassanakajon, 2013). The proPO gene encodes the phenoloxidase precursor enzyme, with L. vannamei encoding LvproPO1 and LvproPO2 (Wang et al., 2010). Interestingly, proPO has been shown to be active during WSSV infection (Abad-Rosales et al., 2019). In this study, proPO gene expression exhibited a significant decrease at 48 h compared with that observed in the control. Several authors have reported that during infection by WSSV, the virions are not covered by melanin due to the participation of at least two viral proteins identified in P. monodon. wsv164 interacts directly with PmproPO1 and PmproPO2 (Sangsuriya et al., 2018), probably for an alternative route of infection. Similarly, the WSSV453 protein interacts with the proPO activating enzyme 2 (PmPPAE2), which could explain the observed decrease in the expression of the proPO gene due to the presence of AgNPs, taking into account that no challenge against WSSV was performed in this experiment (Sutthangkul et al., 2017; Sutthangkul et al., 2017). Taken together, these results suggest that Argovits-4 does not induce the expression in of the assayed early stage immune-related genes, suggesting that the antiviral activity or immunostimulation by AgNP involves a different route to the endocytic pathway. A previous study by our group (Ochoa-Meza et al., 2019) reported that the LGBP and Cu,ZnSOD genes are negatively deregulated by the presence of AgNPs (Argovit®), which suggested that there must be a path for the internalization of the nanoparticles and the activation of the immune system of haemocytes. However, such a path does not require the participation of any of the five genes analysed in this study.

Conclusions

In this study, we explored whether a new AgNP formulation (Argovit-4), which replaced Argovit®, has similar properties to protect and immunostimulant shrimp, allowing them to avoid infection by the white spot virus without causing toxic effects. In addition, we assessed if the expression of some genes immune system-related suggested a common means of activation.

1. Decreased or high concentrations of Argovit-4 nanoparticles do not generate mortality by causing toxicity towards shrimp.

2. The mixture of WSSV-Argovit-4 was capable of decreasing the mortality of shrimp by at least 50% by intramuscular administration.

3. Argovit-4 included in feed could be used to prevent infection by WSSV and can be considered a safe candidate to safeguard aquaculture systems in the future.

4. The zero expression of the PAP, Rab6, PEN4 and Crustin genes, suggest that the antiviral activity of Argovit-4 does not include any of the mechanisms by which such genes perform their function in the immune system of L. vannamei.

5. The late down regulation of proPO gene expression, suggests that the antiviral activity of Argovit-4, could to modify a different and unknown route to inhibit a WSSV infection.

Supplemental Information

Supplemental Information 1 Selection of AGNP use concentration

These data correspond to the selection of the concentration of use for the AgNP.

Click here for additional data file.

Supplemental Information 2 Cumulative mortality of shrimp subjected to the Argovit-4 + WSSV mixture

Bioassay of shrimp submitted to the Argovit-4 + WSSV mixture by intramuscular route.

Click here for additional data file.

Supplemental Information 3 Bioassay to protector effect of Argovit-4 against WSSV

These data represent bioassay determined the effect protector of Argovit-4 in feed against a specific inoculum of WSSV.

Click here for additional data file.

Supplemental Information 4 Relative expression data raw

The data correspond to each of the 5 genes evaluated in their relative-expression in the presence of Argovit-4.

Click here for additional data file.

Supplemental Information 5 Normality test of data expression

The data correspond to the normalization of the relative-expression of each gene (PAP, Rab6, PEN4, Crustin and ProPO) in the presence of Argovit-4.

Click here for additional data file.

For technical support: Rene R. Rebollar Prudente from the Biotechnological Marine Organism Laboratory at the Northwest Biological Research Center (CIBNOR), and Carina Gámez Jimenez, from the Biology and Biochemistry Laboratory of Centro de Investigaciones Interdisciplinarias para el Desarrollo Integral Regional del Instituto Politécnico Nacional (CIIDIR-IPN), for all the facilities and technical support during the experiments.

Additional Information and Declarations

Competing Interests

Author Contributions

Data Availability

The authors declare there are no competing interests.

Carlos R. Romo-Quiñonez conceived and designed the experiments, performed the experiments, prepared figures and/or tables, and approved the final draft.

Ana R. Álvarez-Sánchez analyzed the data, prepared figures and/or tables, authored or reviewed drafts of the paper, funding for open access, and approved the final draft.

Pindaro Álvarez-Ruiz conceived and designed the experiments, performed the experiments, analyzed the data, prepared figures and/or tables, and approved the final draft.

Maria C. Chávez-Sánchez, Nina Bogdanchikova and Alexey Pestryakov analyzed the data, authored or reviewed drafts of the paper, and approved the final draft.

Claudio H. Mejia-Ruiz conceived and designed the experiments, analyzed the data, prepared figures and/or tables, authored or reviewed drafts of the paper, and approved the final draft.

The following information was supplied regarding data availability:

Raw data is available in the Supplemental Files.

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
