# Peer review of "Evaluation of a new Argovit as an antiviral agent included in feed to protect the shrimp Litopenaeus vannamei against White Spot Syndrome Virus infection"

_PeerJ, doi:10.7717/peerj.8446_

## Round 0.1 · original submission · Major Revisions

Dear Authors, thank you for you manuscript. Both reviewers have recommended major revisions for your manuscript. Please note there is no recommendation for further experiments but a broader discussion of the results and experiments in the context of the exposure concentrations. Please also respond to the comments regarding the appropriateness the analysis undertaken.

Reviewer 1 ·

Basic reporting

The manuscript by Quiñonez et al. evaluated a new antiviral agent against the White Spot Syndrome Virus (WSSV) infection in the shrimp Litopenaeus vannamei. The propose was to incorporate Argovit, a formulation containing silver nanoparticles, in the food of L. vannamei as a prophylactic treatment for this important virus disease. The authors evaluated the viability of this proposal by performing several tests.
The article explored the literature regarding the importance and the advantages of using silver nanoparticles (AgNP) as an antiviral agent in those shrimps. Nevertheless, not even one reference was made to the toxicity of silver (Ag) and AgNP to crustaceans or aquatic organisms, despite there is literature reported about it (e.g. Fabrega, J., et al. Environ. Int. v 37, p 517, 2011 and George, et al., ACS Nano v 6, p 3745, 2012). You must consider and describe the disadvantages of the use of this AgNP food as well, especially because it is linked with one of the questions you were trying to answer (whether Argovit-4 supplemented in feed exhibits toxicity towards shrimp).
In general, the work is well written, clear and logically presented. The Figures and Tables presented are relevant to the content of the article and properly described and labelled. Though, all the figures are not presented in a sufficient resolution, particularly Figure 4, in which the legends of the y-axis are almost impossible to read. All the appropriate raw data have been made available.

Experimental design

Methods were described with sufficient information to be reproducible by another investigator. Basically, the authors divided the experimental design in 4 bioassays aiming to achieve 4 different answers.
An experimental concern regarding the “Preliminary Bioassay 1: Toxicity test of Argovit-4 supplemented in feed” need to be addressed. This bioassay was performed to answer whether Argovit-4 supplemented in feed exhibits toxicity towards shrimp. The experimental design for testing the toxicity of Ag as AgNP was conducted to answer only a small piece of this important and complex question. Why did the authors choose to perform the experiment for only 192 hours? It may be not a sufficient period to look at effects on crustaceans after exposure via food. A longer period of exposure would be required to ensure the non-toxic effect of the Argovit-4 supplemented in feed, otherwise, the findings of this work are answering only some acute effects questions. What about the chronic effects?
Additionally, the endpoints chosen (“signs of toxicity”, such as lethargic movement, empty gut or atypical colour change and mortality) are the tip of the iceberg. We do not know, from the results of this experiment, if there are occurring internal and sub-lethal effects (bioaccumulation, distribution of Ag to the tissues, histo and/or physiological alterations). Adding this to the short time of exposure, I am afraid you cannot state that there are no toxic effects of Argovit-4 supplemented in feed to the shrimp.

Validity of the findings

The toxic effects of Ag and its nanoparticles should not be taken as a simple and trivial question, despite the successful results obtained using them against the WSSV. The highest concentration tested, and the most effective against WSSV, was 1000 µg/g. This is an enormous amount of AgNP. Taking this into account, there are some thoughts to be considered in the discussion:
- What would be the satisfactory frequency of feeding with Argovit-4 to achieve the expected results against WSSV infection? Would it be occasional or continuous?
- The observation of AgNP chronic effects should be considered.
- Is there Ag accumulation by the shrimp? If there is Ag accumulation, what would be the impact of this accumulation in shrimp consumption?
- It would improve your findings regarding AgNP toxic effects if you also looked at sub-lethal effects. For example, you already collect the haemolymph to perform the assay with haemocytes. It would be very useful to use this haemolymph also to determine Ag concentration. By doing that, you could investigate internal doses of Ag after feeding and whether this Ag is being absorbed - and distributed throughout the body - or is being excreted (Vannuci-Silva, M. et al. Environ. Toxicol. Chem, vol 38, n. 4, 806. 2019).
- How could we avoid the contamination of the marine environment (water and biota) that surrounds the shrimps’ farms?

Additional comments

The benefits of AgNP as broad-spectrum antimicrobial can be used to solve many issues, however, we must be aware of its possible risks. A deeper discussion about the toxic effects of AgNP on the crustaceans is necessary. The study suggests the use of a substantial amount of AgNP (up to 1000 µg/g) to combat WSSV infection. I strongly recommend other ecotoxicological assays using longer periods of exposure (chronic) and observing also sub-lethal and “non-visible” effects. The possibilities to avoid contamination of the aquatic environment by the waste of AgNP would be valuable information too.

Reviewer 2 ·

Basic reporting

Reviewer comments:

Overall and interesting paper in merit of publication; however, the statistical assessment and description are greatly lacking. Please re-analyse your data using adequate statistical models for the mortality assays.
The introduction and methods are well written, but I think the results and discussion may change once the data are correctly analysed. The use of an ANOVA may be correct but the data types need to be explicitly mentioned. If parametric statistics are required, I suggest modelling the data in a glm or using a basic Kruskal test.

Major comments:
The statistical analysis is really the only part of the process that seems to be lacking. All the experiments seem to be well planned and rigorously executed. You should apply a form of mortality model, such as is available from the coxme package in R. The data type (normal?) is not reported, so it is difficult to know if the correct tests have been applied, but I doubt in this case that they are sufficient to analyse the data. In addition, the result of all statistical tests are not adequately represented in the text. You should be explicit in saying which test was used and the resulting output such as degree of freedom and supporting statistics like r2 or others depending on the test you use and the exact p-value.

Minor comments:
General English needs to be corrected.

There is no mention about whether this antiviral might have negative impacts on humans after consumption? Could this be elaborated upon in the discussion?

Fig 3 – this should be re-plotted using a better statistical model, possibly using the R package coxme and related packages.

Figures: Please keep your colour system the same throughout. You use blue triangles for different controls across graphs and it is confusing.

Experimental design

See report

Validity of the findings

See report

Additional comments

A great paper, but your analysis needs work. I can suggest it is accepted once the analysis is sound.

---

## Round 0.2 · Major Revisions

Dear Authors

Please can you resubmit your rebuttal letter and manuscript so that it more clearly articulates to the editor/reviewers how and where you have made changes based on the initial comments.

I can see that you have made comments based on the reviews but it isn’t clear what you have or haven’t changed in the revised manuscript. Please state where you have made changes and if you haven’t please state why you haven’t if you disagree with the reviewer suggestions.

I will send the manuscript back out to review following the resubmission.

---

## Round 0.3 · accepted · Accept

Dear Authors, thank you for submitting your revised manuscript. I'm delighted to accept your manuscript for publication. Please look out for some of your statistics which indicate a p = 0.0000 and round up to p < 0.0001

Reviewer 1 ·

Basic reporting

A discussion regarding the toxicity of silver (Ag) and AgNP to crustaceans or aquatic organisms was made. Figure 4 was rebuilt and separated into 3 figures, which improved the visualization and comprehension of them.

Experimental design

The authors have addressed the answers regarding bioassay 1 satisfactorily.

Validity of the findings

The authors have addressed the answers regarding the environmental concerns of using AgNP fo shrimp treatment. The questions that could not be solved by this work pointed out for the next studies.

Additional comments

The authors considered the reviewers' comments and suggestions. The questions that could not be answered by this study were pointed out in the discussion and further studies were proposed aiming to solve them.